# Proteomic Landscape Has Revealed Small Rubber Particles Are Crucial Rubber Biosynthetic Machines for Ethylene-Stimulation in Natural Rubber Production

**DOI:** 10.3390/ijms20205082

**Published:** 2019-10-14

**Authors:** Dan Wang, Quanliang Xie, Yong Sun, Zheng Tong, Lili Chang, Li Yu, Xueyan Zhang, Boxuan Yuan, Peng He, Xiang Jin, Yiyang Dong, Hongbin Li, Pascal Montoro, Xuchu Wang

**Affiliations:** 1College of Life Sciences, Key Laboratory for Ecology of Tropical Islands, Ministry of Education, Hainan Normal University, Haikou 571158, China; wangdanqz2009@126.com (D.W.); xiequanliang001@163.com (Q.X.); yulixjnu@163.com (L.Y.); zhangxueyan_caas@126.com (X.Z.); 15830503299@163.com (B.Y.); jinx@hainnu.edu.cn (X.J.); 2Institute of Tropical Biosciences and Biotechnology, Chinese Academy of Tropical Agricultural Sciences, Haikou 571101, China; tongzheng@itbb.org.cn (Z.T.); tychang493875@126.com (L.C.); 3Key Laboratory of Xinjiang Phytomedicine Resource and Utilization of Ministry of Education, College of Life Sciences, Shihezi University, Shihezi 832003, China; lihbabc@163.com; 4Rubber Research Institute, Chinese Academy of Tropical Agricultural Sciences, Danzhou 571737, China; sunyong_03119308@126.com (Y.S.); penghe1974@163.com (P.H.); 5College of Life Science and Technology, Beijing University of Chemical Technology, Beijing 100029, China; yydong@mail.buct.edu.cn; 6CIRAD, UMR AGAP, F-34398 Montpellier, France; pascal.montoro@cirad.fr

**Keywords:** *Hevea brasiliensis*, natural rubber biosynthesis, protein phosphorylation, quantitative proteomics, small rubber particle

## Abstract

Rubber particles are a specific organelle for natural rubber biosynthesis (NRB) and storage. Ethylene can significantly improve rubber latex production by increasing the generation of small rubber particles (SRPs), regulating protein accumulation, and activating many enzyme activities. We conducted a quantitative proteomics study of different SRPs upon ethylene stimulation by differential in-gel electrophoresis (DIGE) and using isobaric tags for relative and absolute quantification (iTRAQ) methods. In DIGE, 79 differentially accumulated proteins (DAPs) were determined as ethylene responsive proteins. Our results show that the abundance of many NRB-related proteins has been sharply induced upon ethylene stimulation. Among them, 23 proteins were identified as rubber elongation factor (REF) and small rubber particle protein (SRPP) family members, including 16 REF and 7 SRPP isoforms. Then, 138 unique phosphorylated peptides, containing 129 phosphorylated amino acids from the 64 REF/SRPP family members, were identified, and most serine and threonine were phosphorylated. Furthermore, we identified 226 DAPs from more than 2000 SRP proteins by iTRAQ. Integrative analysis revealed that almost all NRB-related proteins can be detected in SRPs, and many proteins are positively responsive to ethylene stimulation. These results indicate that ethylene may stimulate latex production by regulating the accumulation of some key proteins. The phosphorylation modification of REF and SRPP isoforms might be crucial for NRB, and SRP may act as a complex natural rubber biosynthetic machine.

## 1. Introduction

The Para rubber tree (*Hevea brasiliensis*) is the only economic plant to produce natural rubber [1,2]. The rubber particle, which is a specific organelle in the rubber latex, is widely considered as the only site for natural rubber biosynthesis (NRB) and storage [3]. Based on their sizes, rubber particles are classified as large rubber particles (LRPs) and small rubber particles (SRPs) [4]. Rubber transferase activity in SRPs is much higher than that in LRPs [5]. Small rubber particle protein (SRPP), a rubber biosynthesis-related protein that accumulates mainly in SRPs, plays a more positive role in rubber biosynthesis [6]. The histochemical localization of the SRPP and rubber elongation factor (REF) indicates that rubber biosynthesis in the rubber laticifer is mostly concentrated in SRPs [7]. In natural rubber production, latex is collected from laticifers by regularly tapping the trunk barks of the Para rubber tree [3,8]. Treatment of the rubber tree bark with ethephon increases the volume of the rubber latex several-fold, and this practice has been widely used in commercial latex production [9,10]. However, it is a longstanding mystery that most key genes in natural rubber biosynthesis are significantly repressed upon the application of ethylene [11,12].

Here, we have performed a comprehensive proteomics study of ethylene-stimulated rubber latex and have identified more than four hundred ethylene responsive proteins from the total rubber latex [12]. We also noticed that specific isoforms of REF and SRPP are important for ethylene-stimulated latex production in rubber latex. Notably, we found that ethylene improves the generation of SRPs in the rubber latex, and proteins in SRPs might play more important roles in the total rubber latex production [12]. Therefore, in this study, we further performed an in-depth proteomics study of SRPs with different ethylene treatments using two quantitative proteomics methods, namely, differential in-gel electrophoresis (DIGE) and using isobaric tags for relative and absolute quantification (iTRAQ). As a gel-based method, DIGE can determine highly abundant proteins on the 2-DE gels. Different to DIGE, iTRAQ is a gel-free high throughput method, and it can usually produce several thousands of unique proteins in many plants. Based on the new data produced by the two complementary proteomics methods, we uncovered some novel mechanisms of ethylene-stimulation in rubber latex production in SRPs.

## 2. Results

### 2.1. Morphological Analysis of Rubber Particles in the Para Rubber Tree

Both the transmission electron microscopy (TEM) (Figure 1A) and scanning electron microscopy (SEM) (Figure 1B–D) results demonstrate that the examined rubber particles are spherical or pear-shaped, with diameters ranging from 50–2500 nm. We determined the diameter of approximately 2600 rubber particles and found 803 SRPs (about 31%, less than 200 nm) and 871 (accounting for 33.5%, larger than 400 nm) LRPs (Figure 1E). We further calculated the abundance of all the detected rubber particles and noticed that only a 4.1% abundance was obtained from the 803 SRPs, whereas the 871 LRPs accounted for 85.2% abundance. The middle rubber particles (200 to 400 nm) occupied 10.7% abundance (Figure 1F). These calculated results, as well as the observations in the TEM and SEM images, revealed that there exist large amounts of SRPs and a high abundance of LRPs in rubber laticifer cells.

### 2.2. Identification of DAPs in SRPs by 2-D DIGE

The SDS-PAGE of proteins from different SRPs showed that the gel background was clear (Appendix A). The global protein patterns on the differential in-gel electrophoresis (DIGE) gels with 1057 ± 58 spots were stable, and 79 DAPs (termed as D1–D79), containing 49 unique proteins, were positively identified by MS (Figure 2 and Appendix A, Appendix A). After treatment with ethylene for 24 h, 59 proteins were significantly up-regulated and only 12 proteins were down-regulated. Prolonging the treatment time to 48 h, less proteins were obviously changed, including 29 increased and 28 decreased proteins. Although most NRB-related proteins were not significantly changed, some protein isoforms or proteoforms were sharply induced by the addition of ethylene at 24 and/or 48 h.

Among the identified DAPs, 23 protein isoforms were REF/SRPP family members, including 16 REF and 7 SRPP isoforms. Except for SRPP243 (spot D22), the other 22 REF/SRPP members were localized in the genome scaffold1222. All of the 23 REF/SRPP isoforms were induced after ethylene treatment for 24 h, and 9 isoforms (spots D-20, 31, 35, 50, 53, 66, 70, 73 and 74) were still induced with the elongation of treatment time, but 10 members were not significantly changed under ethylene treatment for 48 h, and even 4 isoforms (spots D-22-24 and 31) were decreased under ethylene treatment for 48 h. Three unique proteins, named REF138, REF258, and REF175, were identified from the 16 REF isoforms. Four unique proteins, named SRPP117, SRPP154, SRPP204, and SRPP243, were identified as SRPPs from seven protein isoforms (Figure 2I, Appendix A).

### 2.3. Determination of Phosphorylated Peptides in REF and SRPP

Furthermore, 64 REF/SRPP protein spots in different 2-DE gels were further examined, using MS/MS analysis to determine their phosphorylated amino acids. They were positively identified as 23 protein isoforms in the REF/SRPP family. These protein isoforms belong to four unique SRPP/REF family proteins (named REF138, REF175, REF258, and SRPP204). Finally, 344 phosphorylated peptides were identified from different protein isoforms in the REF/SRPP family under different treatments, and they contain 138 unique phosphorylated peptides, which were identified from REF138, REF175, REF258, and SRPP204. Among them, 53, 8, 35, and 42 unique phosphorylated peptides were determined from REF138, REF175, REF258 and SRPP204, respectively. These peptides contain 129 phosphorylated amino acids and they were obtained from the 64 protein spots. These phosphorylated amino acids include 75 serine, 31 threonine, 11 lysine, 8 aspartic acid and 4 tyrosine (Figure 3 and Appendix A, Appendix A).

Six protein spots (S-50-53, 74 and 79) were determined as REF138, containing 53 unique phosphorylated peptides and 27 phosphorylation amino acids. The six protein isoforms were detected from the six spots under four (D24, D48, E24 and E48) treatments, by a total of 24 MS/MS experiments (Appendix A, Appendix A). Six protein isoforms were significantly up-regulated after E24 treatment, and five of them kept increasing with the elongated ethylene treatment (Appendix A). With the elongated treatment, more phosphorylated amino acids were observed and a total of seven unique phosphorylated peptides were identified in the E48 sample. Six phosphorylated amino acids (Asp 4, Asp 6, Ser 43, Ser 81, Lys 128, and Lys 132) were examined in the E48 sample. Except for spot S52, the other spots had more phosphorylated amino acids upon ethylene stimulation, especially for spots 53 and 79 in the E48 treatment (Figure 3A). Ser 81 was phosphorylated under both the E24 and E48 treatments, but it was not a phosphorylated amino acid in D48 (Appendix A). Only 8 phosphorylated peptides with 6 phosphorylated amino acid sites were observed in the two REF175 spots (Figure 3B). However, the phosphorylated amino acids decreased after ethylene stimulation, indicating that the phosphorylation of REF175 might be not as important as in the other REF members.

Five spots (S-20, 21, 23, 24, and 27) were identified as REF258, containing 35 phosphorylated peptides with 16 phosphorylated amino acids that were detected from the five spots under four (D24, D48, E24, and E48) treatments, by a total of 20 MS/MS experiments (Appendix A, Appendix A). They were sharply up-regulated after being treated with ethylene for 24 h, but three spots (S-23, 24 and 27) decreased in the E48 treatment (Appendix A, Appendix A). Seven spots were identified as SRPP, among them, SRPP204 was the most abundant (Appendix A). Three abundant SRPP204 spots, namely, S31, S49, and S70, which were induced by ethylene stimulation, were further selected to detect their phosphorylated peptides. The three SRPP204 isoforms contained 42 phosphorylated peptides with 24 phosphorylated amino acids. These results revealed that large amounts of peptides were phosphorylated in SRPP204, and 24 amino acids were phosphorylated in the detected three SRPP204 isoforms (Figure 3C, Appendix A, Appendix A).

### 2.4. Identification and Determination of DAPs in SRPs by iTRAQ

Two (R1 and R2) independent 8-plex isobaric tags for relative and absolute quantification (iTRAQ) experiments were performed to determine the DAPs in the SRPs from the D24-, D48-, E24-, and E48-treated rubber latex samples. Altogether, 2596 high-quality unique proteins were determined in the SRPs (Appendix A). Venn diagram analysis resulted in 2024 shared unique proteins between R1 (2371 unique proteins) and R2 (2249 unique proteins). There were 347 and 225 specific proteins in R1 and R2, respectively (Figure 4A). Furthermore, we detected 226 DAPs (named as I1 to I226) from the 2024 shared unique proteins in SRPs. These DAPs changed at least 2-fold (*p* < 0.01) upon ethylene stimulation (E24 and/or E48). Among them, 174 proteins were up-regulated and 126 proteins were down-regulated after ethylene treatment for 24 and/or 48 h. Ten shared DAPs were detected from the iTRAQ and DIGE experiments (Figure 4A).

Based on the changed patterns, the 226 DAPs from iTRAQ were classified into four areas. The first area contained 60 proteins, which were reduced under E24 but induced in E48. The second area contained 22 proteins which were induced under all ethylene treatments. The third area contained 15 decreased proteins under all ethylene treatments. Finally, the fourth area contained 129 proteins which were induced under E24 but reduced in E48. Our interest was focused on the 22 proteins in the second area that were induced under all ethylene treatments. The 22 proteins in the second area mainly contained REF175 (I66), Nogo-B receptor (I69), rubber cis-poly prenyltransferase HRT2 (I116), elongation factor 1-gamma (I22), albumin-2 like protein, and the vacuolar protein sorting-associated protein 27 (Appendix A).

### 2.5. Functional Analysis of DAPs in SRPs

We further performed gene ontology (GO) distribution analysis using the WEGO software package, analyzing the 305 DAPs (including the 213 up-regulated and 146 down-regulated proteins, which contain 265 unique proteins) obtained from the iTRAQ and DIGE analysis. The enriched outputs were 213 up-regulated and 146 down-regulated proteins (Figure 4, Appendix A). At the molecular function level, nine subcategories were enriched. Among them, the largest portion, including 71 up-regulated and 34 down-regulated proteins, is involved in organic cyclic compound binding (Appendix A). Fifteen components were observed at the subcellular level. Among them, most proteins are localized to being intracellular, intracellular part, intracellular organelle, or intracellular organelle part. There were also many proteins involved in the membrane-bound organelle, protein complex, endomembrane system, or the intrinsic component of membrane (Figure 4D; Appendix A). For the biological process, 14 main pathways were detected. Among them, the largest portion, including 130 proteins, was involved in the organic substance metabolic process. A large part, including 71 proteins, was related to the single-organism metabolic process (Figure 4E, Appendix A).

Clustering of the 46 sharply changed proteins (those with at least a 5-fold change) upon ethylene stimulation revealed that the protein changed patterns under E24/D24 and D48/D24 were similar, while E48/D48 and E48/E24 showed a similar change ratio for protein accumulation in the SRPs. In E24/D24, 18 proteins were sharply induced. After ethylene treatment for 48 h, only four proteins were sharply induced, and seven proteins were decreased (Figure 5A, Appendix A).

### 2.6. Integrative Analyses of DAPs in Different SRPs

We further performed western blotting to determine the general accumulation patterns of 9 typical proteins in SRPs (Figure 5B) and calculated their relatively changed ratios (Figure 5C). Among them, 8 proteins could be clearly detected in D24, where only the accumulation of the ethylene-inducible protein (EIP) was relatively low. In E24, six proteins, namely, SRPP, *Hevea* rubber cis-prenyltransferase (HRT), EIP, plasma membrane intrinsic protein (PMIP), heat shock protein 70 (HSP70), and acetyl-CoA carboxylase (ACOC) were significantly induced, and three proteins, named REF, ascorbate peroxidase (APX) and glutathione reductase (GLR), were not changed significantly. In D48, the abundance of almost all proteins (except for REF) was significantly up-regulated, but half of them were decreased under E48 treatment (Figure 5C). SRPP and REF, as the two most abundant proteins, showed differently changed patterns upon ethylene stimulation in SRPs. The E24 and D48 treatments could significantly increase the accumulation of SRPP, but decreased the abundance of REF. Western blotting confirmed that ethylene stimulation at both 24 and 48 h could sharply induce the accumulation of HRT. EIP was significantly up-regulated by the ethylene and D48 treatments, and PMIP was mainly accumulated under the E24 and D48 treatments. HSP70 could be induced by all of the three treatments. APX and GLR showed similarly changed patterns, and they were rich in E48, but had a low abundance in E24. ACOC was also significantly enhanced by ethylene stimulation in SRPs (Figure 5C, Appendix A).

Given the results from this SRPs proteomics study and the recent published literature, we summarized the main functions of the identified proteins and their potential locations in rubber latex, proposing a possible schematic representation of the intrinsically proteome-based mechanism for the possible regulation details of proteins in SRPs (Figure 6). These results reveal that the ethylene stimulation of rubber latex production performs on the protein accumulation level in rubber particles, especially in SRPs. The post-translational modification of some key proteins and many metabolic enzymes might play crucial roles in the control of NRB process in SRPs.

## 3. Discussion

### 3.1. Proteome Profiling Revealed SRPs Contain Almost all NRB-Related Proteins

Many proteins involved in the NRB process have been identified by proteomics-based technologies from different latex components [12,13,14,15,16,17,18,19]. However, both the identified number of proteins and the quantification of target proteins are limited, probably due to the lack of assembled high-quality rubber tree genomes and high-resolution mass spectrometry data. Recently, more proteomics studies of rubber particles have been performed, and more NRB-related enzymes, including AACT, hydroxymethylglutaryl coenzyme A synthase (HMGS), mevalonate kinase (MEVK), farnesyl pyrophosphate/diphosphate synthase (FADS), geranylgeranyl pyrophosphate synthase (GGPS), etc., have been identified as membrane-bound/attached proteins in purified rubber particles [16,17,18,19,20,21]. In a previous proteomic study, 186 proteins were positively identified from rubber particles, containing several isoforms of REF, SRPP, and cis-prenyl transferase (CPT) [20]. After the removal of the abundant REF and SRPP bands, 137 proteins, including 115 unique proteins, were further identified [18]. The protein profiles of LRPs and SRPs resulted in 53 DAPs, containing 22 gene products, such as SRPP, REF, HMGS, HSP70, and phospholipase D [16]. Interaction network analysis of rubber particle proteomics has revealed that the protein complexes of HRT1, REF, and HRT1-REF bridging protein (HRBP) might play crucial roles as NRB machinery [18], and these proteins are associated with the endoplasmic reticulum [22].

Our 2-DE gel-based proteomics study resulted in more than 100 abundant proteins from rubber particles [23,24], which is similar with previous reports [16,18,20]. In the past decade, only 313 proteins were identified from the Para rubber tree, and 1208 proteins from all rubber-producing plants [25]. Recently, more than 1600 proteins were identified from rubber total latex by iTRAQ [12], and 1839 unique proteins were determined by LC MS/MS from the whole translated draft *Hevea* genome [19]. Our SRPs proteomics data produced more than 2000 unique proteins (Figure 4A, Appendix A), which covers almost all the previously identified proteins. Pathway analysis (Figure 6) revealed that almost all proteins involved in NRB could be detected in SRPs.

### 3.2. Ethylene Improved NRB by the Accumulation of Rubber Producing-Related Proteins in SRPs 

Ethylene stimulation can sharply induce natural rubber production, but the expression level of most key genes in NRB, such as FADS [3], CPT [26] and hydroxymethylglutaryl coenzyme A reductase (HMGR) [27], are not significantly up-regulated [11,12]. At the protein level, new evidence supports the accelerative effect of ethylene on NRB [28,29]. The accumulation of several proteins, including glutamine synthetase [30], phosphoenolpyruvate carboxylase [17], acetyl-CoA C-acetyltransferase [12,17], sucrose transporter [31], and aquaporin [32], are activated upon ethylene application. At the proteomic level, ethylene induces the accumulation of many enzymes involved in carbohydrate metabolism, energy production, and NRB [12,17]. Here, we determined 305 ethylene responsive proteins from DIGE (Appendix A) and iTRAQ (Appendix A) and found that half of them were induced by ethylene stimulation (Figure 6).

NRB is a typical isoprenoid metabolic process [33,34], beginning with isopentenyl pyrophosphate (IPP) synthesis in the menvalonate (MVA) pathway [7,35]. In early steps, acetyl-CoA C-acetyltransferase (ACAT) is important for generating acetoacytyl-CoA, where it catalyzes acetyl-CoA to form acetoacetyl-CoA, which is the first step in the MVA pathway. Then, HMGS and HMGR activate the supply of mevalonate substrates [12,33]. Our study showed that, ACOC, as an important enzyme for the biosynthesis of fatty acids, was mainly accumulated in SRPs (Appendix A), but could not be detected in the total rubber latex [12]. Through a series of enzymatic reactions, acetyl-CoA can be catalyzed to form IPP [34]. Three ACAT genes were determined in the rubber tree [7]. Our previous proteomic results revealed that both the gene and protein accumulation of ACAT are depressed upon ethylene stimulation, but one protein isoform is induced [12]. In this study, one isoform (D12) was identified as ACAT from SRPs by DIGE and iTRAQ experiments. Its abundance was induced after E24 treatment but depressed under E48 treatment (Appendix A), which is consistent with our previously published latex proteomics results [12].

HMGS is the second enzyme in the MVA pathway [29,36]. Here, two HMGS members were found in SRPs. HMG-CoA is further converted to MVA by HMGR. However, we did not detect HMGR, probably due to its low abundance in SRPs. MEVK controls the first stage in the conversion of mevalonate into isopentyl pyrophosphate. It has a higher accumulation in rubber latex and may be coded by a single gene [7]. However, three genes were determined in a recently annotated genome of the rubber tree [8]. In this proteomics study, we also identified five MEVK members in SRPs (Appendix A), indicating that more MEVK members are involved in NRB after the application of ethylene. Mevalonate disphosphate decarboxylase (MEVD) catalyzes MVAPP to form IPP [35]. It was reported to be coded by a single gene [7], but genomic analysis resulted in two MEVD genes [8], and in this study, we also obtained two MEVD members in the SRPs (Appendix A).

In the natural rubber elongation process, geranyl pyrophosphate synthase (GPS) catalyzes IPP and dimethylallyl diphosphate (DMAPP) to form geranyl diphosphate (GPP), and FADS catalyzes farnesyl diphosphate (FPP) to form polyisoprene by adding IPP [2,37]. GGPS and geranylgeranyl pyrophosphate synthase (GGPPS) catalyze farnesyl diphosphate to form geranylgeranyl diphosphate (GGPP) [18,33]. We identified GPS, FADS, and GGPPS in SRPs and determined a FADS member (I48) as an ethylene responsive protein (Appendix A).

### 3.3. Different Proteoforms in SRPs May Play Different Roles in NRB

Genomic analysis demonstrated that most proteins involved in MVA pathways are encoded by multiple genes, meeting the formation of different isoprenoids in the Para rubber tree [8,38,39]. HRT or CPT is a key enzyme in natural rubber biosynthesis, and it uses pyrophosphates to initiate rubber molecules [36,40]. It is a rubber particle membrane protein [18]. A total of eleven HRT members have been determined in the recently published rubber tree genome [8], but only HRT2 shows HRTase activity [2]. Its activity is higher in SRPs than in LRPs [21]. At the gene level, the expression of CPT1 and CPT2 are both depressed by ethylene [11,12]. However, our previous results revealed ethylene significantly improved HRT2 abundance in the rubber latex [12]. Here, we further proved HRT2 abundance (I116) was significantly induced by ethylene in SRPs (Figure 5C, Appendix A), indicating HRT2 plays a more important role in ethylene-induced latex production in SRPs.

During the elongation process of rubber biosynthesis, REF, SRPP, and a Nogo-B receptor HRBP are known as RP membrane binding proteins, playing a role as a natural rubber biosynthetic machinery [18,41]. We identified HRBP (I69) in SRPs and found that it could be significantly induced by ethylene (Appendix A). REF and SRPP are closely combined with rubber particles [42]. A recent report indicates that SRPP can recruit CPT to the endoplasmic reticulum to interact with HRBP [22]. In *Hevea*, 18 REF/SRPP family members are observed, and their genes exhibit distinct expression patterns in different tissues; most genes are located in Scaffold1222 in a single 205-kb genome site [8,15]. In 2-DE gels, we determined 28 protein isoforms from five REF/SRPP members, and multiple protein isoforms were identified [15]. Generally, the gene expression and protein accumulation for these REF/SRPP members is decreased or not changed after ethylene treatment. However, individual REF/SRPP isoforms display differently changed patterns upon ethylene stimulation in different rubber tree clones [8,15].

In this SRPs proteomics study, 23 REF/SRPP isoforms were identified as DAPs upon ethylene application, and they were induced by ethylene treatment for 24 h (Appendix A). Among them, many amino acids were phosphorylated upon ethylene stimulation (Appendix A). It is widely known that both phosphorylation and dephosphorylation are predominant intracellular posttranslational modification mechanisms, playing important roles in different cell-signaling pathways [43]. Two key classes of enzymes that are essential in controlling the opposite action of these posttranslational modifications are kinases and phosphatases (responsible for phosphorylation and dephosphorylation, respectively), which correspondingly initiate and terminate cellular signals. Our phosphoproteomics analysis of the rubber latex has revealed that many phosphorylated proteins are substantially more apparent after ethylene treatment, and a large amount of ethylene-responsive proteins with phosphatase/phosphotransferase activity and kinase activity are enriched [15]. We also found that the abundance of some latex proteins in the rubber tree were significantly reduced after ethylene stimulation [15]. Recently, Habib et al. collected proteins from the C-serum, lutoids, and rubber particle layers of the rubber latex, determining 73 post-translationally modified peptides in the 47 identified rubber latex proteins, including the main PTMs, such as phosphorylation, lysine acetylation, and ubiquitination [43].

In this study, we noticed that four isoforms of the identified 23 REF/SRPP members were decreased after ethylene stimulation for 48 h (Figure 2I, Appendix A), and the phosphorylated amino acids were decreased (Figure 3B; Appendix A), indicating that the dephosphorylation of proteins might also be important in NRB. These results are consistent with the recent reports in total latex [15,44], indicating ethylene stimulation mainly occurs in SRPs and phosphorylation, and/or that the dephosphorylation of different proteoforms in SRPs might play different crucial roles in NRB.

### 3.4. Comprehensive Proteomics Analysis Revealed SRP Might Be an Efficient Machine for NRB

In the rubber tree, cis-1,4-polyisoprene is generated through the MVA pathway, involving initiation, elongation, and termination [7,33,35]. The enzymes for polyisoprene initiation include ACAT, HMGS, HMGR, MEVK, and phosphomevalonate kinase (PMVK). During the elongation process, REF, SRPP, CPT (HRT), GPS, GGPPS, FADS and HRBP are key proteins, and they were all identified from SRPs (Appendix A), and most of them were induced after ethylene application (Figure 6). These proteins and cofactors may formulate a natural living carbocationic polymerization (NLCP) mechanism for carotenoid biosynthesis [33]. The proteomics study of rubber particles and interaction network analyses revealed a protein complex consisting of HRT, REF, and HRBP, which function as rubber biosynthetic machinery in the rubber particles [18]. Our results herein demonstrated that SRP with enzymes should be a more active NLCP machine for NRB.

In a recent proteomics study, we have noticed that ethylene can improve SRPs generation and prolongs the latex flowing time [12]. Here, we observed that, the general abundance of pro-hevein was not significantly changed after ethylene treatment (Appendix A), but six isoforms of pro-hevein were determined as ethylene-responsive proteins on DIGE gels (Figure 2). Among them, five isoforms were significantly depressed by E24 treatment, but four isoforms were induced under E48 treatment (Figure 6). Hevein and its precursor pro-hevein are traditionally considered as major proteins in lutoids, they can inhibit rubber particle aggregation and latex coagulation [3,14,45]. The decreased accumulation of hevein in latex maintains the flow of latex and finally improves latex yield after ethylene application [12].

Here, many SRP proteins were directly or indirectly involved in NRB. We identified 24 members of peptidyl-prolyl cis-trans isomerase (PP isomerase) from SRPs, but only one member (I176) was induced by ethylene stimulation (Appendix A), indicating that this isoform may be more important than others in SRPs. In the SRPs, we also identified an albumin-2 like protein (I12), and it was sharply induced by ethylene stimulation (Appendix A). This protein may contribute to osmotic regulation in laticiferous cells upon ethylene stimulation. A universal stress protein was also previously observed in total rubber particles (TRPs) [20]. In SRPs, its two members were sharply accumulated after ethylene treatment for 24 h but decreased in the E48 treatment (Figure 6).

Methyltransferase can transfer a methyl group from S-adenosylmethionine to their substrates. This has been identified from the rubber latex [12,17,19] and C-serum [46]. Its abundance was significantly up-regulated upon the application of ethylene to the total rubber latex [12] and SRPs (Appendix A). Vacuolar protein sorting-associated protein (VPSP) has been detected from rubber latex [47]. Here, we found VPSP4, as well as REF175, elongation factor 1 and ribosomal protein L18-2, were sharply up-regulated after ethylene treatment for 24 h in SRPs (Figure 6).

Given our in-depth proteomics data in SRPs and the results from the recent literature, we proposed an intrinsically subcellular-based mechanism associated with both the localizations and responses of major SRP proteins after ethylene stimulation. We suggested a general scheme of the ethylene-induced biochemical pathways in SRP at the protein level (Figure 6). This new schematic model considers SRP as a natural rubber biosynthetic machine in response of ethylene stimulation. The newly obtained proteomics data revealed that SRPs contain almost all proteins involved in NRB, and many proteoforms are induced by ethylene. The changed patterns of some proteoforms are different from the observations of that in the gene expression level [11,12] and general protein accumulation patterns [12,17]. In total rubber latex, ethylene stimulation not only increases the accumulation of most NRB-related enzymes, but also induces many biochemical processes in SRPs. On the contrary, enzymes involved in latex coagulation can be inhibited upon ethylene application, which depresses rubber particle aggregation and the formation of rubber coagulates in SRPs and subsequently prolongs the latex flowing time. As a feedback mechanism, natural rubber biosynthesis can be enhanced and some proteoforms of key rubber biosynthesis proteins, particularly REF and SRPP, are activated by phosphorylation. We considered that phosphorylation and/or dephosphorylation on some specific isoforms of key enzymes in SRPs may be a key regulation factor for ethylene stimulation of latex yield. SRPs can play a role as an efficient machine for ethylene-stimulation of natural rubber production (Figure 6). This is the first in-depth quantitative proteomics study of SRPs upon ethylene treatment, and the first report on the role of the small rubber particle as an efficient machine for NRB.

## 4. Materials and Methods

### 4.1. Plant Material and Ethylene Treatment

A total of 60 regularly tapped (~10-year-old) rubber trees (*H. brasiliensis Mull. Arg*., clone RY 7-33-97) were selected and randomized into four groups, containing three biological replicates. These matured plants were not tapped for one month, then applied with 3% ethephon, as described [12,13]. An equal volume of ddH_2_O was used as a control. One day (24 h) later, these plants were tapped to collect fresh latex. After 2 days (48 h), these plants were tapped for the second time. After tapping, the first ten drops were discarded. The subsequent latex samples were collected in ice-chilled glass beakers for further study. The latex samples collected from these plants treated with ddH_2_O (control) and 3% ethephon for 24 and 48 h were termed as D24, D48, E24 and E48, respectively. Five technical replicates were performed for each treatment.

### 4.2. Preparation of Total Rubber Particles and Purification of SRPs

The TRPs were prepared as previously described [23]. In brief, the ice-chilled rubber latex was centrifuged (Beckman Model JA 2-21, Beckman Coulter, Fullerton, CA, USA) with 40,000 *g* for 60 min at 4 °C. After ultracentrifugation, 3 major layers were observed. The top white zone was total rubber particles, the middle aqueous layer was C-serum, and the bottom fraction was termed as lutoids. The upper cream of rubber particles was collected into a new tube, and then, were further suspended in an ice-cold washing solution (WS) (20 mM Tris-HCl, 300 mM mannitol, 0.5 mM DTT, pH 7.0) in ration of 1:10 (*w/v*). These mixtures were gently stirred for 30 min and then centrifuged at 30,000 *g* for 15 min at 4 °C. Repeat the above washing steps twice. Finally, the floated phrase was collected as total rubber particles and the non-rubber fractions were discarded.

Then, TRPs were separated into LRPs and SRPs by a percoll-medium-based density gradient centrifugation method as described [48]. In brief, a percoll medium (1.31 mg/mL, Cat. 17-0891-01, Pharmacia) was diluted in WS contained 5%, 15% and 30% percoll (*v/v*) to produce series of different separation solutions, respectively. The collected total rubber particles were suspended in the WS contained 30% percoll in ration of 1:10 (*w/v*). These mixtures have the highest density (termed as 30% separation media) and they were then transferred into a new tube (10 mL) as the bottom layer. Then, this layer was carefully covered with the WS contained 15% percoll and finally capped with the WS contained 5% percoll as the top layer. The gradients were centrifuged with 5000 *g* for 15 min at 4 °C. After centrifugation, the white mixtures in the top one third of the tube were collected and washed once with WS, then centrifuged with 5000 *g* for additional 15 min at 4 °C. The top layer was collected as LRPs fraction. At the same time, the bottom one third of the tube were collected and washed once with WS, then centrifuged with 5000 *g* for additional 15 min at 4 °C. The bottom layer was collected as SRPs fraction. Finally, total rubber particles were separated into large rubber particles and small rubber particles by this newly developed percoll medium density gradient centrifugation protocol.

### 4.3. Morphological Analysis of Large and Small Rubber Particles

TEM and SEM were used to examine the ultrastructural patterns of rubber particles. TEM analysis was performed as described [14]. At the same time, the purified TRPs, SRPs, and LRPs were examined under SEM as described [4,12]. The collected TRPs, LRPs and SRPs were mounted on aluminum stubs, coated with gold particles, and examined under an S-3000 SEM (Hitachi, Tokyo, Japan).

### 4.4. Protein Extraction, 2-D DIGE and MALDI TOF/TOF MS Analysis

The proteins from SRPs were extracted as described [12,23]. DIGE analysis was performed to determine DAPs as described [12]. Proteins of SRPs from the D24, D48, E24, and E48 latex were labeled with a ratio of 400 pmol Cy3 or Cy5. Cy2 was used to label the internal standard produced by pooling an equal protein quantity from the four samples. Differential in-gel analysis module was used for spot detection, and biological variation analysis module was employed in the three biological replicates to identify DAPs with higher than 99% confidence (*p* < 0.01, *t*-test value < 0.05). Furthermore, these DAPs were manually excised and in-gel digested with bovine trypsin as described [12,15]. Mass spectra of the peptides were acquired on an AB 5800 MALDI-TOF/TOF MS instrument as described [12,15]. These peptide masses were searched against the *Hevea* genome scaffolds (BioProject ID: PRJNA80191, www.ncbi.nlm.nih.gov/nuccore/448814761) and the draft genome (GenBank: AJJZ01000000) with 46,718 sequences and 17,435,757 residues [8]. Positive identification was based on a Mascot score higher than 75 (*p* < 0.001), at least 2 matched peptides, and a maximum peptide coverage higher than 5%. Detailed information about protein identification is provided in Appendix A.

### 4.5. Identification of Phosphopeptide by MS/MS

The in-gel digestion of the protein spots was performed as described [12,15]. The digested peptides were collected and the phosphopeptides were enriched by Titansphere Phos-Tio Kit (GL Science, Tokyo, Japan) following the manufacturer’s instruction. The enriched phosphopeptides were determined by a Triple TOF 6600 system (AB SCIEX). The identified results were accepted if they have a greater than 95% peptide probability and contained at least two identifiable phosphopeptides. Only the phosphopeptides with Ascore localization probability minimum of 99% were considered as high probability. The protein function motif was analyzed by the MEME Suite V4.12.0 (http://meme-suite.org/index.html), and 3D structure modeling and the potential phosphorylated sites of these proteins were built through CPH models 3.2 Server (http://www.cbs.dtu.dk/services/CPHmodels).

### 4.6. Labeling and Quantification of SRPs Proteins by iTRAQ

Two independent iTRAQ experiments were carried out on the Triple TOF 6600 system (AB SCIEX). The vacuum dried peptides were labeled with the eight iTRAQ tags named 113 (D24), 114 (E24), 115 (D48), 116 (E48), 117 (D24-2), 118 (E24-2), 119 (D48-2) and 121 (E48-2), respectively. The MS analysis was performed in Information Dependent Mode. Protein identification from the rubber tree genome as described above and relative iTRAQ quantification was performed with ProteinPilot™ Software 5.0 (AB SCIEX). A strict unused confidence score >2.0 was used. The identified proteins with at least two matched peptides, confidence higher than 95%, and an FDR value ≤ 1% were used to perform protein quantification. Subsequently, proteins with 2.0-fold change (*p* < 0.01) were termed as DAPs in SRPs from the four biological replicates.

### 4.7. Protein Classification and Hierarchical Cluster Analysis

The identified proteins were searched against the UniProt (http://www.ebi.uniprot.org/) database to confirm their functions. Proteins were then classified using Functional Catalogue software (http://mips.gsf.de/projects/funcat) to obtain their COG codes. GO pathway analysis was performed by Blast2-GO software (Valencia, Spain) [14].

### 4.8. Western Blotting Analysis

Western blotting was performed with three biological replicates as described [14]. The changed ratios of these proteins in the different SRPs were calculated on band abundance in Western blotting by ImageQuant TL software (GE Healthcare, Uppsala, Sweden).

## Figures and Tables

**Figure 1 ijms-20-05082-f001:**
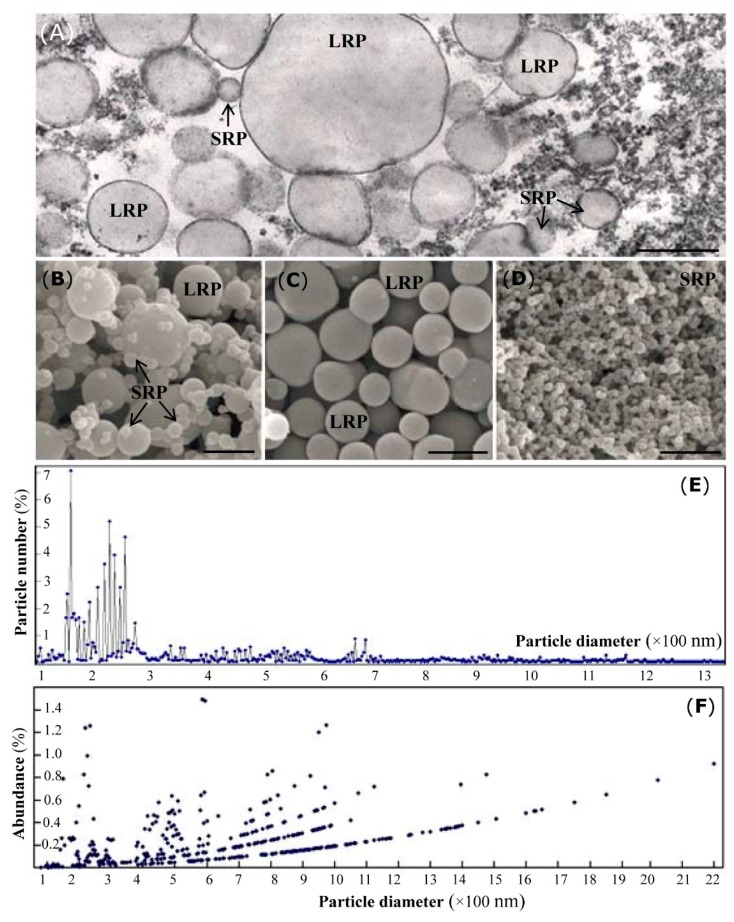
Morphological patterns of the Para rubber particles. Typical ultrastructural patterns of large rubber particles (LRPs) and small rubber particles (SRPs) were examined under TEM (**A**) and SEM (**B**–**D**), respectively. Under TEM, many spherical and pear-like LRPs were observed (A). Under SEM, more SRPs were detected in the total rubber particles (TRPs) collected from the fresh rubber latex (**B**). After separation, no SRPs were detected from the purified LRPs (**C**), and vice versa (**D**). The percent number (**E**) and abundances (**F**) of different rubber particles are highlighted. Bar, 1.0 μm.

**Figure 2 ijms-20-05082-f002:**
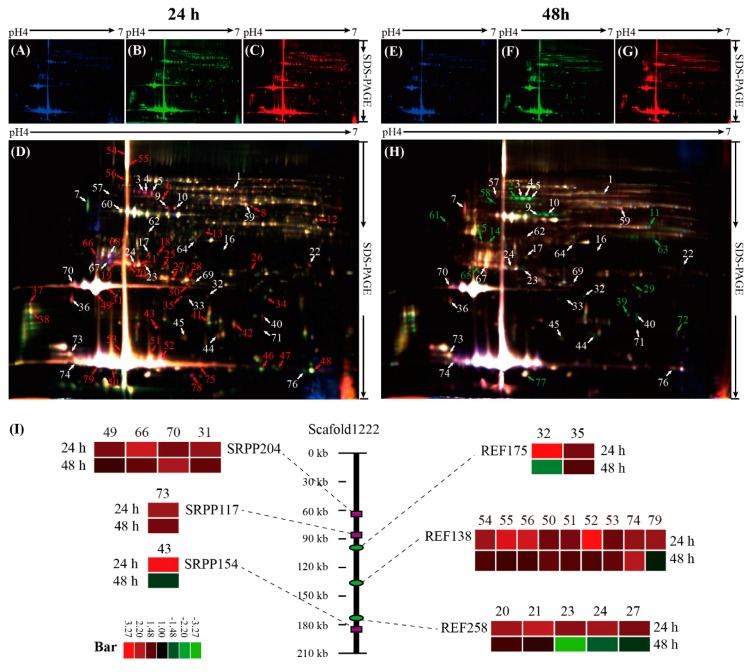
Differential in-gel electrophoresis (DIGE) analysis of SRP proteins upon ethylene stimulation. Internal standards (IS) for protein mixtures are labeled with Cy2 and shown in blue color (**A**,**E**); Cy3, green, for D24 (**B**) and D48 (**F**); Cy5, red, for E24 (**C**) and E48 (**G**). The combined images of D24, E24, and IS (**D**), and IS with D48 and E48 (**H**) are highlighted. The 79 DAPs (from D1 to D79) are numbered in different colors (red, up-regulated; green, down-regulated) and their identifications are listed in Appendix A. The scaffold locus and changed patterns of the rubber elongation factor (REF) and small rubber particle protein (SRPP) family members are presented (**I**).

**Figure 3 ijms-20-05082-f003:**
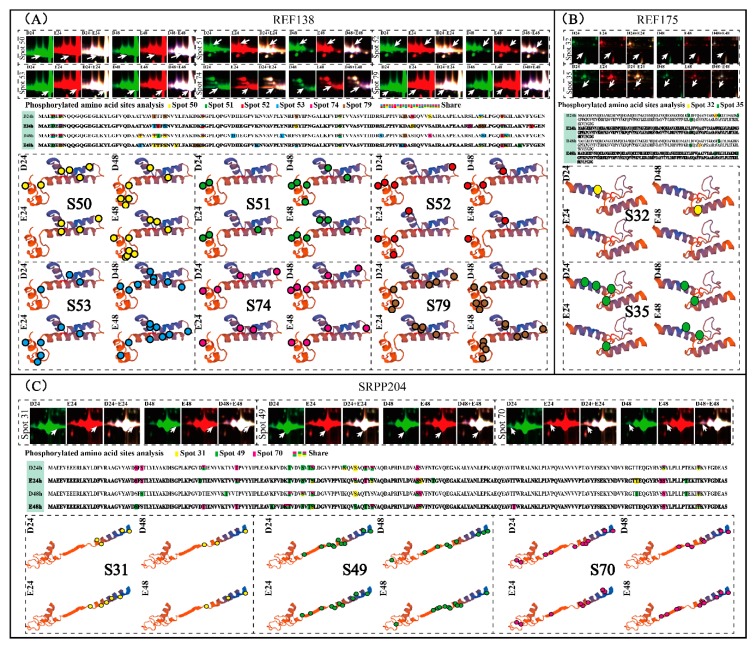
Phosphopeptides and phosphorylated amino acids in different REF and SRPP members. The changed pattern of different protein spots after the D24, D48, E24, and E48 treatments on the DIGE gels, phosphorylated peptides, phosphorylated amino acid sites, and the predicted 3-D structures for REF/SRPP family members REF138 (**A**), REF175 (**B**), SRPP204 (**C**), and REF258 (Appendix A) are highlighted. The capital letters and circles with different color represent the phosphorylated amino acids in different protein spots. The detail information for these peptides and phosphorylated amino acid sites is listed in Appendix A and Appendix A.

**Figure 4 ijms-20-05082-f004:**
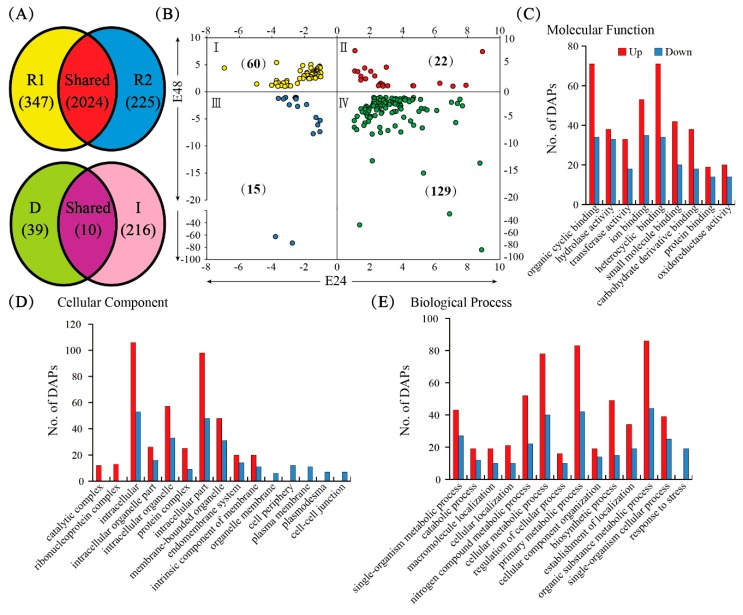
Isobaric tags for relative and absolute quantification (iTRAQ) analysis of SRPs proteins upon ethylene stimulation. The Venn diagram demonstrates that 2024 unique proteins were shared in the two iTRAQ experiments (**A**). A total of 226 DAPs were determined from iTRAQ, and 49 unique proteins from 2-D DIGE, which resulted in 10 shared unique proteins (**A**). The DAPs from iTRAQ were classified into four areas based on their changed patterns in different SRPs (**B**). The statistical protein numbers of the 265 DAPs involved in molecular function (**C**), the cellular components, (**D**) and biological process (**E**) are highlighted.

**Figure 5 ijms-20-05082-f005:**
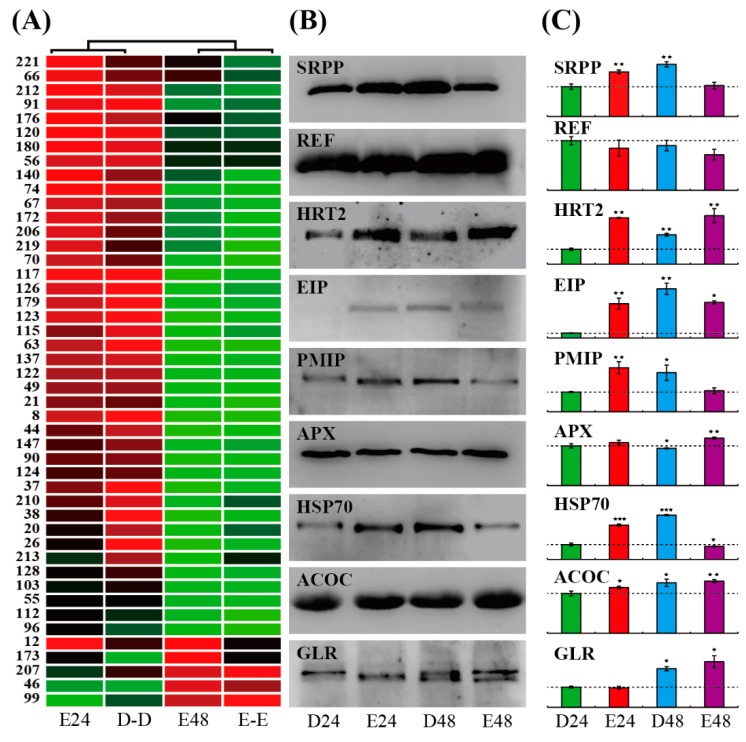
Accumulation patterns of the typical ethylene-responsive proteins in SRPs. The 4 columns represent the 46 sharply changed proteins under the E24/D24, D48/D24, E48/D48, and E48/E24 treatments (marked as E24, D-D, E48, and E-E). The up- or down-accumulated proteins are indicated in red or green color, respectively. The protein number from iTRAQ (I-) is indicated on left side (**A**). Western blotting showed the general protein accumulation patterns (**B**). The dotted line represents the changed ratio value of 1.0 (**C**). * *p*, 0.05, ** *p*, 0.01, and *** *p*, 0.001.

**Figure 6 ijms-20-05082-f006:**
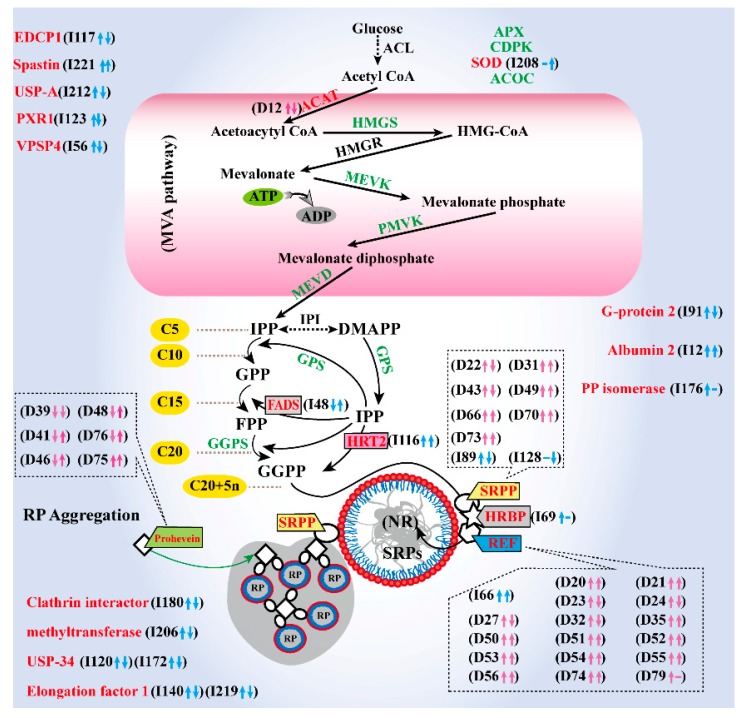
Schematic diagram of SRP proteins involved in natural rubber biosynthesis (NRB). A schematic diagram of enzyme localization and the regulation of ethylene-induced biochemical pathways on the protein level in SRPs are both highlighted. In this schematic diagram, the identified proteins involved in NRB from SRPs are marked with different colors (green, unchanged protein; red, changed protein), and the changed patterns determined by 2-D DIGE (D-) and iTRAQ (I-) for these DAPs are provided in the parentheses. The up arrow stands for upregulation, the down arrow stands for downregulation, and minus sign the protein is not significantly changed after ethylene application.

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
