# Peer review of "Proteomic Landscape Has Revealed Small Rubber Particles Are Crucial Rubber Biosynthetic Machines for Ethylene-Stimulation in Natural Rubber Production"

_ijms, 2019, doi:10.3390/ijms20205082_

Round 1

Reviewer 1 Report

This study describes proteomic analysis of small rubber particles (SRPs) to identify proteins and phosphoproteins influenced by Ethephon treatment, which induces increase in latex volume. Authors used DIGE and iTRAQ labeling strategy for quantitative and differential proteomics and reported differentially expressed proteins and phosphoproteins. They claimed identifying many proteins related to ethylene stimulation and involved in rubber biosynthetic pathways. I have several comments concerns about the manuscript, which needs to be addressed before this can be published.

Major:

In the title “simple or complex?” seems nothing to do with the whole study. It’s confusing. Can be simply deleted and keep the rest. The paper is mostly descriptive and I understand this is a typical for any proteomics paper. But, I think authors attempt to making extensive biological interpretation from these global and differential proteomics data, and the results could be hardly connected to such detail biological interpretation.

I don’t understand the reason for combining DIGE with iTRAQ labeling data. DIGE samples were analyzed by MALDI TOF/TOF MS and iTRAQ by Triple TOF 6600 system. Why such a small overlap between DIGE and iTRAQ data despite significantly large coverage of proteins by iTRAQ? This raises some questions about the identification of DAP, particularly by DIGE. Only 10 (20%) of the 49 DAP in DIGE were also identified as DAP in iTRAQ despite identifying 226 DAP (Fig. 4A).

I also have a concern about phosphoproteomics. 565 phosphorylated peptides containing 577 phosphorylated amino acids were obtained from the 64 protein spots including 383 serine, 112 threonine, 52 aspartic acids, 26 lysine and 4 tyrosine. That seems too many phosphopeptides for such a small number of proteins. How did authors confirm that they can detect lysine and aspartic acid phosphorylation? My understanding is that mapping these phosphorylated residues require special buffers and enrichment condition. This requires detail explanation how these phosphorylation sites were identified and validated. Are those 565 phosphorylated peptides unique peptides or also include repeated counts of the same peptide. Did author used peptide extracted from each spot to do phosphopeptide enrichment?

In line 126: authors mentioned that six protein spots (S-50-53, 74 and 79) were determined as REF138, with 208 phosphorylated peptides and 27 phosphorylation amino acids. This is unclear. Does that mean, authors are reporting 208 phosphorylated peptides from six proteins. That means 34 phosphopeptides/protein but only 27 phosphorylation amino acid. I am totally lost here.

The same question arises in the para starting in line 138: Five spots (S-20, 21, 23, 24 and 27) were identified as REF258, containing 213 phosphorylated peptides with 16 amino acids. How is it possible to identify so many phosphopeptides from such a small number of protein.

I saw a similar paper by the authors published in Scientific Reports in 2015 (Sci Rep. 2015, 5:13778). In that paper, they used the same DIGE and iTRAQ method to compare proteome of ethylene stimulation. I don't know how this paper differs from that published work in scientific aspect. Any explanation how current work differs from that work would be helpful.

Minor:

Protein species: Avoid using “protein species.

Abstract: REF first mentioned in abstract in abbreviated form. Provide expanded form for the first time.

Line 49: add that after “indicated”

Line 95: change “After treated” to “After treating”

Line 95: Significantly increased to significantly up-regulated

Line 139: treated to treating

Line 221: Give the obtained result to given the results from this………

Line 440: Change the target protein spot performed in-gel digestion to “the in-gel digestion of the protein spots”

Author Response

Response to Reviewer One

Comment 1: This study describes proteomic analysis of small rubber particles (SRPs) to identify proteins and phosphoproteins influenced by Ethephon treatment, which induces increase in latex volume. Authors used DIGE and iTRAQ labeling strategy for quantitative and differential proteomics and reported differentially expressed proteins and phosphoproteins. They claimed identifying many proteins related to ethylene stimulation and involved in rubber biosynthetic pathways. I have several comments concerns about the manuscript, which needs to be addressed before this can be published.

Response: Thank you very much for this Reviewer to give us so many detail corrections and helpful suggestions. According to the Reviewer’s suggestions, the authors have further carefully revised this manuscript. The pointed mistakes have been corrected in the revised version. The English language of the revised manuscript was carefully polished by all the authors to minimize typographical, grammatical, and bibliographic errors. Some typographical errors, grammar, spelling, vocabulary, and confusing sentences were corrected. Therefore, this newly revised version might be more suitable for publication. We have also submitted a highlighted version showing where revisions have been made with point-by-point replies to your previously raised concerns of the original manuscript.

Comment 2: In the title “simple or complex?” seems nothing to do with the whole study. It’s confusing. Can be simply deleted and keep the rest.

Response: We agree with this Reviewer that the original tittle is confusing, and based on this Reviewer’s suggestion, we have deleted “simple or complex?” and provided a new tittle named “Proteomic landscape revealed small rubber particle is a crucial rubber biosynthetic machine for ethylene-stimulation of natural rubber production” in the revised version.

Comment 3: The paper is mostly descriptive and I understand this is a typical for any proteomics paper. But, I think authors attempt to making extensive biological interpretation from these global and differential proteomics data, and the results could be hardly connected to such detail biological interpretation.

Response: We agree with this Reviewer that, as a typical paper for proteomics, we mainly discussed the possible roles of these proteins that have been identified by MS from this study. Furthermore, we performed Western blotting to determine the general accumulation patterns of 9 typical proteins in SRPs and calculated their relatively changed ratios. Our Western blotting results also confirmed that ethylene stimulation at both 24 and 48 hours could sharply induce the accumulation of many proteins in SRPs. Finally, combined our SRPs proteomics data and the recent published literatures, we summarized the main functions of the identified proteins and their potential locations in rubber latex, and proposed a possible schematic representation of the intrinsically proteome-based mechanism for the possible regulation details of proteins in SRPs. These results revealed that ethylene stimulation of rubber latex production performs on protein accumulation level in rubber particles, especially in SRPs. Therefore, we think our interpretation is mainly based on our SRPs proteomics data and the conclusions are acceptable for a proteomics-based study.

Comment 4: I don’t understand the reason for combining DIGE with iTRAQ labeling data. DIGE samples were analyzed by MALDI TOF/TOF MS and iTRAQ by Triple TOF 6600 system. Why such a small overlap between DIGE and iTRAQ data despite significantly large coverage of proteins by iTRAQ? This raises some questions about the identification of DAP, particularly by DIGE. Only 10 (20%) of the 49 DAP in DIGE were also identified as DAP in iTRAQ despite identifying 226 DAP (Fig. 4A).

Response: In this study, we detected differentially accumulated SRP proteins by using two comparative proteomics methods named DIGE and iTRAQ. DIGE is a gel-based method, and can usually be used to determine the high abundant protein on the 2-DE gels. As a gel-based method, DIGE can produce about one thousand protein spots on the gel and it is widely used to detect the DAPs that produced by post-modification of unique protein. The 2-DE gel is the only visual method to determine the DAPs from different post-modifications, and this kind of DAPs are usually named as protein isoforms or protein isoforms in traditional proteomics study. As a high throughput gel-free method, iTRAQ can detect several thousands of unique proteins and usually generate several hundreds of DAPs. Therefore, DIGE and iTRAQ are two complementary proteomics methods. In this proteomics study, we have conducted both qualitative and quantitative proteomics analyses of SRPs. In qualitative proteomics analysis, we provided the protein profiles of SRPs in both SDS-PAGE (Figure S1) and 2-D DIGE gels (Figure 2). In the DIGE gels, we determined 79 DAPs (D1-D76) as ethylene responsive proteins, which contain only 49 unique proteins. Among the 79 DAPs, 23 protein isoforms were identified as REF/SRPP family members, including 16 REF and 7 SRPP isoforms. Then, the total proteins in SRPs were determined by two (R1 and R2) independent high throughput iTRAQ method, and finally identified 2,024 shared unique proteins between R1 (2,371 unique proteins) and R2 (2,249 unique proteins) in SRPs. We also detected 226 DAPs (named as I1 to I226) from the 2,024 shared unique proteins in SRPs by iTRAQ method. Therefore, DIGE and iTRAQ are two different kinds of quantitative proteomics and usually produced a small overlap in many published proteomics papers. In this study, we detected ten shared DAPs from iTRAQ and DIGE experiments (Figure 4A).

Comment 5: I also have a concern about phosphoproteomics. 565 phosphorylated peptides containing 577 phosphorylated amino acids were obtained from the 64 protein spots including 383 serine, 112 threonine, 52 aspartic acids, 26 lysine and 4 tyrosine. That seems too many phosphopeptides for such a small number of proteins. How did authors confirm that they can detect lysine and aspartic acid phosphorylation? My understanding is that mapping these phosphorylated residues require special buffers and enrichment condition. This requires detail explanation how these phosphorylation sites were identified and validated. Are those 565 phosphorylated peptides unique peptides or also include repeated counts of the same peptide. Did author used peptide extracted from each spot to do phosphopeptide enrichment?

Response: In this proteomics study, we selected a total of 64 REF/SRPP isoforms in different 2-DE gels to performed MS/MS analysis to determine their phosphorylated amino acids (Figure 2, Figure S2, Table S1). They only belong to four unique SRPP/REF family proteins, named REF138, REF175, REF258 and SRPP204. Finally, 565 phosphorylated peptides containing 577 phosphorylated amino acids were obtained from the 64 protein spots; they include 383 serine, 112 threonine, 52 aspartic acid, 26 lysine and 4 tyrosine (Figure 3, Figure S3, Table S2). Six protein spots (S-50-53, 74 and 79) were determined as REF138, with 208 phosphorylated peptides and 27 phosphorylation amino acids (Figure S3). Five spots (S-20, 21, 23, 24 and 27) were identified as REF258, containing 213 phosphorylated peptides with 16 amino acids. Seven spots were identified as SRPP; among them, SRPP204 is the most abundance one (Table S1). The three SRPP204 isoforms contain 132 phosphorylated peptides with 24 amino acids. These results revealed that large amounts of peptides were phosphorylated in SRPP204, and 24 amino acids were phosphorylated in the detected three SRPP204 isoforms (Figure 3C, Figure S3, Table S2).

Therefore, we really detected 565 phosphorylated peptides containing 577 phosphorylated amino acids from the 64 protein spots that belonged to only four unique SRPP/REF family proteins, and their detail information can be obtained from the Supplementary Materials Figure S3 (Phosphopeptides and phosphorylation amino acid sites in different REF and SRPP isoforms) and Table S2 (Phosphopeptides of phosphorylation amino acid sites in SRPP and REF isoforms). From the Supplementary Materials Figure S3 and Table S2, we can find the detail information on the phosphopeptides for these proteins. The detail phosphorylation of the 26 lysine (K 26 Lys) and 52 aspartic acids (D 52 Asp) can be obtained from Table S2.

We completely agree with this Reviewer that mapping these phosphorylated residues require special buffers and enrichment condition. In the Materials and Methods section 4.5 (Identification of phosphopeptide by MS/MS), we have provided the detail steps on how these phosphorylation sites were identified and validated. We have declared that the digested peptides were collected and the phosphopeptides were enriched by Titansphere Phos-Tio Kit (GL Science, Tokyo, Japan) following the manufacturer’s instruction. The enriched phosphopeptides were determined by a Triple TOF 6600 system (AB SCIEX). The identified results were accepted if they have a greater than 95% peptide probability and contained at least two identifiable phosphopeptides.

From the Supplementary Materials Figure S3 and Table S2, we can also know that the 565 phosphorylated peptides are not unique peptides and they also include repeated counts of the same peptide detected from different protein spots.

Comment 6: In line 126: authors mentioned that six protein spots (S-50-53, 74 and 79) were determined as REF138, with 208 phosphorylated peptides and 27 phosphorylation amino acids. This is unclear. Does that mean, authors are reporting 208 phosphorylated peptides from six proteins. That means 34 phosphopeptides/protein but only 27 phosphorylation amino acid. I am totally lost here. The same question arises in the para starting in line 138: Five spots (S-20, 21, 23, 24 and 27) were identified as REF258, containing 213 phosphorylated peptides with 16 amino acids. How is it possible to identify so many phosphopeptides from such a small number of protein.

Response: In this study, REF138 were identified from six protein spots (Spots 50, 53, 74 and 79). From Supplementary Table S2 and Figure S3, we know that there are only a total of 138 amino acids in REF138, and after in-gel digested by bovine trypsin, only a limited number of peptides can be produced from this protein. The 208 phosphorylated peptides were the total number collected from the six protein spots (Spots 50, 51, 52, 53, 74 and 79) under the four treatments (D24, E24, D48 and E48). Therefore, a total of 24 times of MS/MS experiments were performed to determine the detail phosphorylated peptides in REF138. These phosphorylated peptides are not unique peptides and they also include repeated counts of the same peptide detected from different protein spots under different treatments. Our results also revealed that only 27 out of the 138 amino acids can be phosphorylated modification under at least one of the four (D24, E24, D48 and E48) treatments.

We agree with this Reviewer that it is difficult to understand by the readers and we have provided more information on the determination of phosphorylated peptides. The revised part were read as: Six protein spots (S-50-53, 74 and 79) were determined as REF138, with 208 phosphorylated peptides and 27 phosphorylation amino acids that were detected from the six spots under the four (D24, D48, E24 and E48) treatment conditions by a total of 24 MS/MS experiments (Figure S3, Table S2).

A similar revision has also been performed for the identification of phosphorylated peptides and amino acids in REF258 from the five protein spots (Spots 20, 21, 23, 24 and 27) in the newly revised manuscript.

Comment 7: I saw a similar paper by the authors published in Scientific Reports in 2015 (Sci Rep. 2015, 5:13778). In that paper, they used the same DIGE and iTRAQ method to compare proteome of ethylene stimulation. I don't know how this paper differs from that published work in scientific aspect. Any explanation how current work differs from that work would be helpful.

Response: Yes, we really published a proteomics paper titled “Comprehensive Proteomics Analysis of Laticifer Latex Reveals New Insights into Ethylene Stimulation of Natural Rubber Production” in Scientific Reports (Sci. Rep. 2015, 5:13778) by using DIGE and iTRAQ methods to compare proteome of ethylene stimulation on the rubber latex. In the published paper, we used the total rubber latex as material, and this study is further conducted based on the main findings in the published paper. As declared in the introduction section of this manuscript, we have performed a comprehensive proteomics analysis of ethylene-stimulated rubber latex, and identified more than four hundred of ethylene responsive latex proteins. In the published paper, we noticed that specific isoforms of REF and SRPP are important for ethylene-stimulation of the total rubber latex production. Notably, we also found that ethylene improves the generation of small rubber particles (SRPs) in the rubber latex and consider that proteins in SRPs might play more important roles in rubber latex production [Wang et al. 2015, Sci. Rep. 5, 13778].

In the published study, we identified 59 differential phosphoproteins in a phosphoproteomics analysis of the total rubber latex after treated with ethylene, and found that, although many phosphorylated proteins were visualized by Pro-Q Diamond for latex under the control condition, the phosphorylated proteins were substantially more apparent after ethylene treatment. When the differentially expressed proteins recovered from DIGE and iTRAQ methods were combined, we noticed a large amount of ethylene responsive proteins with phosphatase/phosphotransferase activity (33 proteins) and kinase activity (22 proteins). Ten proteins showed phosphate-binding activity, and 7 proteins had strong phosphopyruvate hydratase activity. Among these phosphoproteins, REF was the most abundant, followed by the 7 isoforms of SRPP. A large portion, including phosphoesterase, phosphoglycerate kinase, phosphoglucomutase, phospholipase, and 14-3-3 protein, had high phosphotransferase and kinase activity. These enzymes were involved in Ca2+ binding and signal transduction. We also noticed that specific isoforms of rubber elongation factor and small rubber particle protein that can be phosphorylated mainly at serine residues after ethylene stimulation. This post-translational modification and isoform-specific phosphorylation might be important for ethylene-stimulated latex production (Wang et al. 2015, Sci. Rep. 5, 13778). These results indicated that phosphorylation of proteins might be important in the ethylene-stimulation of rubber latex and suggests that rubber latex may contain many phosphorylated enzymes in the small rubber particles.

Therefore, in this study, we performed an in-depth quantitative proteomics of SRPs upon different ethylene treatments, and finally uncovered some novel mechanisms on ethylene-stimulation of rubber latex production.

Comment 9: Protein species: Avoid using “protein species”.

Response: In proteomics study, we usually used the terminology “protein species” to describe the unique protein, which was identified from different spots on the 2-DE gels. In some cases, "protein species" instead of protein can better define what we want to say.  "Gene product" could be also used. Other terms such as isoforms, allelic variants are utilized, but in the context of proteomics could be not correct, not properly used. Have a look to Schluter et al. 2009. Chemistry Centr-al Journal 3: 11. Based on the description of Schluter et al. (Schluter et al. Finding one's way in proteomics: a protein species nomenclature. 2009, Chemistry Central Journal 3: 11), we had changed some terminologies such as protein isoforms or allelic variants to protein species in the original manuscript. But, as this Reviewer has pointed out that, protein species may be not acceptable by the common readers. Therefore, we replaced the terminology “protein species” in with “protein isoforms” in the revised manuscript.

Comment 10: Abstract: REF first mentioned in abstract in abbreviated form. Provide expanded form for the first time.

Response: OK. We have provided all the abbreviations after the Conflicts of Interest section. Based on the reviewer’s suggestion, we have corrected in Line 28, and have been provided the full name of REF, SRPP and other abbreviations when they are mentioned for the first time in the revised manuscript.

Comment 11: Line 49: add that after “indicated”. Line 95: change “After treated” to “After treating”. Line 95: Significantly increased to significantly up-regulated. Line 139: treated to treating.

Response: OK. We have added “that” after “indicated” in Line 49, replaced “After treated” with “After treating” in Line 95, Line 139, and other places, and changed increased to up-regulated in Line 95 and other places in the revised manuscript.

Comment 12: Line 221: Give the obtained result to Given the results from this………Line 440: Change the target protein spot performed in-gel digestion to “The in-gel digestion of the protein spots….”.

Response: OK. We have replaced the words "Given the obtained result" to “Given the results from this” on line 221, and have changed the sentence "the target protein spot performed in-gel digestion" to “The in-gel digestion of the protein spots was performed…” in Line 440 in the revised manuscript.

Reviewer 2 Report

This manuscript is about a comprehensive proteomics analysis of ethylene-stimulated rubber latex. It is important for academic soundness as well as commercial interests that the mechanism of natural rubber biosynthesis is revealed. In the manuscript, the authors performed several types of approaches to obtain the proteins related to rubber biosynthesis based on ethylene stimulation. For its publication to the IJMS, the reviewer requests some points or questions as follows.

1. The iTRAQ method should be mentioned in the Introduction. The method shows a defect to tend to detect proteins associated with membranes. Can it provide the comprehensive proteomics related to rubber biosynthesis?

2. The authors mentioned only HRT2 shows an activity among the HRT family. On the other hand, there are some reports in which HRT1 has a role in the biosynthesis, although it is necessary for HRT1 to prepare a strict condition. Why did the authors assay the activity of all proteins of the family?

3. In Figure 5, are E24 and D48 similar in their up-regulated proteins? What is the reason of the similarity? According to the figure, only HRT2 is regulated on the basis of ethylene stimulation. How about the roles of the other proteins?

4. In Figure 6, can ethylene promote the metabolism of the biosynthesis? In the diagram, several proteins/enzymes seem to have no or less influences from ethylene. In the title, the authors use “rubber biosynthetic machine.” The machine may work along with the pathway. From the present research, it is difficult that the mechanism of the pathway is not revealed.

5. The first appearances of abbreviated terms should be put with the complete names of the terms. For example, in Line 27, REF and SRPP are appeared for the first time.

Author Response

Response to Reviewer Two

Comment 1: This manuscript is about a comprehensive proteomics analysis of ethylene-stimulated rubber latex. It is important for academic soundness as well as commercial interests that the mechanism of natural rubber biosynthesis is revealed. In the manuscript, the authors performed several types of approaches to obtain the proteins related to rubber biosynthesis based on ethylene stimulation. For its publication to the IJMS, the reviewer requests some points or questions as follows.

Response: Thank you very much for this Reviewer to review this manuscript and give us so many helpful suggestions. According to the Reviewer’s suggestions, the authors have further carefully revised this manuscript. The pointed mistakes have been corrected in the revised version. The English language of the revised manuscript was carefully polished by all the authors. Therefore, this newly revised version might be more suitable for publication.

Comment 2: The iTRAQ method should be mentioned in the Introduction. The method shows a defect to tend to detect proteins associated with membranes. Can it provide the comprehensive proteomics related to rubber biosynthesis?

Response: Based on this Reviewer’s suggestion, we have mentioned both the DIGE and iTRAQ methods in the revised Introduction section. The changed part is: Therefore, in this study, we performed an in-depth proteomics of SRPs upon different ethylene treatments by using two quantitative proteomics methods named differential in-gel electrophoresis (DIGE) and isobaric tags for relative and absolute quantification (iTRAQ). DIGE is a gel-based method, and it can determine high abundant proteins on the 2-DE gels. iTRAQ is a gel-free high throughput method, and it can usually generate several thousands of unique proteins in plants. Based on the new data produced by the two complementary proteomics methods, we have uncovered some novel mechanisms on the ethylene-stimulation of rubber latex production.

We totally agree with this Reviewer that rubber particle is a kind of membrane-rich organelle, and we should choose an efficient method to extract proteins from the membrane systems. The protein extraction method used in this study is developed from our Borax/PVPP/Phenol (BPP) method that could efficiently extract proteins from recalcitrant plants [Wang XC, Li XF, Deng X, Han HP, Shi WL, Li YX: A protein extraction method compatible with proteomic analysis for euhalophyte Salicornia europaea L. Electrophoresis 2007, 28:3976-3987] and chloroplast [Fan PX, Wang XC, Kuang TY, Li YX: An efficient method for protein extraction of chloroplast protein compatible for 2-DE and MS analysis. Electrophoresis 2009, 30:3024-3033]. Based on the BPP method, we have developed a new protocol for preparation proteins from the different fractions of rubber latex, especially from the rubber particles. The comparison of our results with the former studies and the identification of some rubber latex specific proteins suggested our method could be more efficient for 2-DE and MS in future latex proteomic analysis [Wang, X.C.; Shi, M.J.; Lu, X.L.; Ma, R.F.; Wu, C.G.; Guo, A.P.; Peng, M.; Tian, W.M. A method for protein extraction from different subcellular fractions of laticifer latex in Hevea brasiliensis compatible with 2-DE and MS. Proteome Sci. 2010, 8, 35]. And this new protocol has been widely used to isolation proteins from different fractions of the rubber latex and generated many proteomics results [Wang, X. C.; Wang, D.; Sun, Y.; Yang, Q.; Chang, L. L.; Wang, L. M.; Meng, X. R.; Huang, Q. X.; Jin, X.; Tong Z. Comprehensive Proteomics Analysis of Laticifer Latex Reveals New Insights into Ethylene Stimulation of Natural Rubber Production. Sci. Rep. 2015, 5, 13778; Wang, X. C.; Shi, M. J.; Wang, D.; Chen, Y. Y.; Cai, F. G.; Zhang, S. X.; Wang, L. M.; Tong, Z.; Tian, W. M. Comparative proteomics of primary and secondary lutoids reveals that chitinase and glucanase play a crucial combined role in rubber particle aggregation in Hevea brasiliensis. J. Proteome Res. 2013, 12, 5146-59; Tong, Z.; Wang, D.; Sun, Y.; Yang, Q.; Meng, X.R.; Wang, L.M.; Feng, W.Q.; Li, L.; Wurtele, E.S.; Wang, X.C. Comparative proteomics of rubber latex revealed multiple protein species of REF/SRPP family respond diversely to ethylene stimulation among different rubber tree clones. Int. J. Mol. Sci. 2017, 18, 958]. Therefore, the method used by us can detect proteins associated with membranes, and it can also provide the comprehensive proteomics related to natural rubber biosynthesis.

Comment 3: The authors mentioned only HRT2 shows an activity among the HRT family. On the other hand, there are some reports in which HRT1 has a role in the biosynthesis, although it is necessary for HRT1 to prepare a strict condition. Why did the authors assay the activity of all proteins of the family?

Response: In the natural rubber biosynthesis pathway, GGPP (Geranylgeranyl diphosphate) can generate natural rubber hydrocarbons contain different length of carbon with the help of cis-prenyl transferase (CPT), which is recently termed as Hevea rubber transferase (HRTase or HRT). HRT is a key enzyme in the natural rubber biosynthesis and can use pyrophosphates to initiate rubber molecule. It is a rubber particle membrane protein [Yamashita et al. 2016. Identification and reconstitution of the rubber biosynthetic machinery on rubber particles from Hevea brasiliensis. ELife, 5, e19022]. A total of eleven HRT members have been determined in the recently published rubber tree genome, but only HRT2 shows HRTase activity [Lau et al. 2016. The Rubber Tree Genome Shows Expansion of Gene Family Associated with Rubber Biosynthesis. Sci. Rep., 6, 28594]. The activity of HRT is higher in SRPs than in LRPs [Rojruthai et al. 2010. In vitro synthesis of high molecular weight rubber by Hevea small rubber particles. J. Bioscie. Bioeng., 109, 107-114].

We agree with this Reviewer’s point that there are some reports in which HRT1 has a role in the biosynthesis, although it is necessary for HRT1 to prepare a strict condition. Another interaction network analysis of rubber particle proteomics has also revealed the protein complex of HRT1, REF, and HRBP might play crucial roles as a NRB machinery. During the elongation process of rubber biosynthesis, REF, SRPP and a Nogo-B receptor named as HRT1-REF bridging protein (HRBP) are known as RP membrane binding proteins that play crucial roles [Yamashita et al. 2016. Identification and reconstitution of the rubber biosynthetic machinery on rubber particles from Hevea brasiliensis. ELife, 5, e19022]. These proteins are associated with the endoplasmic reticulum [Brown et al. 2017. Subcellular localization and interactions among rubber particle proteins from Hevea brasiliensis. J. Exp. Bot., 68, 5045-5055].

However, our previous results revealed ethylene significantly improved HRT2 abundance in the rubber latex [Wang et al. 2015. Comprehensive Proteomics Analysis of Laticifer Latex Reveals New Insights into Ethylene Stimulation of Natural Rubber Production. Sci. Rep., 5, 13778].

In this SRPs proteomics study, we further proved that HRT2 abundance (I116) was significantly induced by ethylene in SRPs (Figure 5C, Table S3). In this proteomics-based study, we did not assay the activity of all the 11 members in the HRT family. Our SRPs proteomics data indicated that HRT2 may play more important roles for natural rubber biosynthesis than the other HRT members in the ethylene-induced latex production in SRPs.

Comment 4: In Figure 5, are E24 and D48 similar in their up-regulated proteins? What is the reason of the similarity? According to the figure, only HRT2 is regulated on the basis of ethylene stimulation. How about the roles of the other proteins?

Response: Thanks for your suggestions. In order to compare the changed patterns for different proteins under different treatments, we selected 46 sharply changed proteins with at least 5.0-fold upon ethylene stimulation. Clustering of the protein changed patterns revealed the E24/D24 and D48/D24 were similar, and E48/D48 and E48/E24 showed similar changed ratios for protein accumulation in SRPs. In E24/D24, 18 proteins were sharply induced. After ethylene treated for 48 hours, only four proteins were sharply induced, and seven proteins were decreased (Figure 5A, Table S3).

We further performed Western blotting to determine the general accumulation patterns of 9 typical proteins in SRPs (Figure 5B) and calculated their relatively changed ratios (Figure 5C). In E24, six proteins named SRPP, HRT, EIP, PMIP, HSP70, and ACOC were significantly induced, and three proteins (REF, APX, and GLR) did not change significantly. In D48, the abundance of almost all proteins (except for REF) was significantly up-regulated, but half of them were decreased under E48 treatment (Figure 5C). E24 and D48 treatments could significantly increase the accumulation of SRPP, but decrease the abundance of REF. Western blotting confirmed that ethylene stimulation at both 24 and 48 hours could sharply induce the accumulation of HRT. EIP was significantly up-regulated by ethylene and D48 treatments, and PMIP was mainly accumulated under E24 and D48 treatments (Figure 5C, Table S3).

Therefore, our Western blotting results of the general accumulation patterns of 9 typical proteins in SRPs (Figure 5B,C) really demonstrated that E24 and D48 show similar effects in their up-regulation of the 9 selected proteins. The reason is not clear, and it is probably due to E24 and D48 treatments can produce similar stimulator for natural rubber biosynthesis.

As we have mentioned that, eleven HRT members have been determined in the recently published rubber tree genome, but only HRT2 shows HRTase activity [Lau et al. 2016. Sci. Rep., 6, 28594]. The activity of HRT is higher in SRPs than in LRPs [Rojruthai et al. 2010. J. Bioscie. Bioeng., 109, 107-114]. Our previous latex proteomics results revealed that ethylene can significantly improve HRT2 abundance in the rubber latex [Wang et al. 2015. Sci. Rep., 5, 13778]. In this SRPs proteomics study, we further proved that HRT2 abundance (I116) was significantly induced by ethylene in SRPs (Figure 5C, Table S3). In this proteomics-based study, we did not assay the activity of all the 11 members in the HRT family, and did not discuss the roles of the other HRT protein members.

Comment 5: In Figure 6, can ethylene promote the metabolism of the biosynthesis? In the diagram, several proteins/enzymes seem to have no or less influences from ethylene. In the title, the authors use “rubber biosynthetic machine.” The machine may work along with the pathway. From the present research, it is difficult that the mechanism of the pathway is not revealed.

Response: We agree with this Reviewer that the detail regulation mechanism in the rubber biosynthetic machine of SRPs is not clearly revealed by this proteomics study. Based on the results from this SRPs proteomics and the recent published literatures, we just summarized the main functions of the identified proteins and their potential locations in rubber latex, and only proposed a possible schematic representation of the intrinsically proteome-based mechanism for the possible regulations of proteins in SRPs (Figure 6). Our SRPs proteomics data produced more than two thousand unique proteins (Figure 4A, Table S3), which covers almost all the previously identified proteins from the rubber latex. Pathway analysis revealed that almost all proteins involved in NRB could be detected in SRPs. Therefore, our proteomics data really revealed that SRPs contain almost all NRB-related proteins and enzymes. Our proteomics data also showed that ethylene can improve NRB by accumulation of rubber producing-related proteins in SRPs. Post-translational modifications of some key proteins and many metabolic enzymes might play crucial roles in controlling NRB in SRPs.

Therefore, in this diagram, we just concluded an intrinsically subcellular-based mechanism associated with both the localizations and responses of main SRP proteins after ethylene stimulation. We also just suggested a possible general scheme of the ethylene-induced biochemical pathways in SRP at the protein level (Figure 6). This new schematic model considers SRPs as a natural rubber biosynthetic machine in response of ethylene stimulation. The newly obtained proteomics data revealed that SRPs contain almost all proteins involved in NRB, and many proteoforms are induced by ethylene. The changed patterns of some proteoforms are different from the observations in the gene expression level and general protein accumulation patterns. We considered that phosphorylation and/or dephosphorylation on some specific isoforms of key enzymes in SRPs may be a key regulation factor for ethylene stimulation of latex yield.

Comment 6: The first appearances of abbreviated terms should be put with the complete names of the terms. For example, in Line 27, REF and SRPP are appeared for the first time.

Response: Thank you for your helpful suggestion. We have provided all the abbreviations after the Conflicts of Interest section. Based on this Reviewer’s suggestion, we have corrected in Line 27 for REF and SRPP, and have been provided the full name of REF, SRPP and other abbreviations when they are mentioned for the first time in the revised manuscript.

Reviewer 3 Report

This manuscript investigates the mechanism of ethylene-stimulation of rubber production. Although the usage of ethylene stimulation of natural rubber, the mechanism is still unclear. Thus the work described here is timely. Minor comments from me:

Supplementary figures were not supplied with the manuscript. Line 120, not sure where do you derive the 64 REF/SRPP proteins Please justify why the DIGE  data deserves to be in the manuscript as the unique proteins found by DIGE is minor compared to iTraq method. Please compare/discuss your work on the phosphorylation of rubber proteins with a recently published related work: Habib, M. A. H., Gan, C. Y., Abdul Latiff, A., & Ismail, M. N. (2018). Unrestrictive identification of post-translational modifications in Hevea brasiliensis latex. Biochemistry and Cell Biology96(6), 818-824.

Author Response

Response to Reviewer Three

Comment 1: This manuscript investigates the mechanism of ethylene-stimulation of rubber production. Although the usage of ethylene stimulation of natural rubber, the mechanism is still unclear. Thus the work described here is timely. Minor comments from me: 1. Supplementary figures were not supplied with the manuscript.

Response: Thank you very much for this Reviewer to review this manuscript. According to this Reviewer’s helpful suggestions, we have further carefully revised this manuscript. The pointed mistakes have been corrected in the revised version. We are very sorry for this mistake. We have added the Supplementary Figures S1-S3 into the new Supplementary Materials in the newly submitted version. These Supplementary Materials will be published online with the publication of this manuscript.

Comment 2: Line 120, not sure where do you derive the 64 REF/SRPP proteins. Please justify why the DIGE data deserves to be in the manuscript as the unique proteins found by DIGE is minor compared to iTraq method.

Response: The description of the 64 REF/SRPP proteins is not proper, we only selected 64 REF/SRPP protein spots in the different 2-DE gels (including D24, D48, E24 and E48 treatments) to perform MS/MS analysis to determine their phosphorylated peptides and phosphorylated amino acids in REF and SRPP proteins. They were positively identified as 23 protein species in the REF/SRPP family. These protein species belong to four unique SRPP/REF family proteins (named REF138, REF175, REF258 and SRPP204) (Figure 2, Figure S2, Table S1). We have changed this paragraph in the revised manuscript.

We also noticed that there is a small overlap of the identified DAPs between DIGE and iTRAQ data. Only 10 (20%) of the 49 DAPs in DIGE were also identified as DAPs in the 226 DAPs from iTRAQ method (Fig. 4A).

In this study, we detected differentially accumulated SRP proteins by using two comparative proteomics methods named DIGE and iTRAQ. DIGE is a gel-based method, and can usually be used to determine the high abundant protein on the 2-DE gels. As a gel-based method, DIGE can produce about one thousand protein spots on the gel and it is widely used to detect the DAPs that produced by post-modification of unique protein. The 2-DE gel is the only visual method to determine the DAPs from different post-modifications, and this kind of DAPs are usually named as protein isoforms or protein species in traditional proteomics study. As a high throughput gel-free method, iTRAQ can detect several thousands of unique proteins and usually generate several hundreds of DAPs. Therefore, DIGE and iTRAQ are two complementary proteomics methods. In this proteomics study, we have conducted both qualitative and quantitative proteomics analyses of SRPs. In qualitative proteomics analysis, we provided the protein profiles of SRPs in both SDS-PAGE (Figure S1) and 2-D DIGE gels (Figure 2). In the DIGE gels, we determined 79 DAPs (D1-D76) as ethylene responsive proteins, which contain only 49 unique proteins. Among the 79 DAPs, 23 protein species were identified as REF/SRPP family members, including 16 REF and 7 SRPP species. Then, the total proteins in SRPs were determined by two (R1 and R2) independent high throughput iTRAQ method, and finally identified 2,024 shared unique proteins between R1 (2,371 unique proteins) and R2 (2,249 unique proteins) in SRPs. We also detected 226 DAPs (named as I1 to I226) from the 2,024 shared unique proteins in SRPs by iTRAQ method. Therefore, DIGE and iTRAQ are two different kinds of quantitative proteomics and usually produced a small overlap in many published proteomics papers. In this study, we detected ten shared DAPs from iTRAQ and DIGE experiments (Figure 4A).

In the Introduction section of the revised manuscript, we have provided more background information for iTRAQ and DIGE methods. The newly added background information may help the potential readers to understand the two methods. The changed part is: Therefore, in this study, we performed an in-depth proteomics of SRPs upon different ethylene treatments by using two quantitative proteomics methods named differential in-gel electrophoresis (DIGE) and isobaric tags for relative and absolute quantification (iTRAQ). DIGE is a gel-based method, and it can determine high abundant proteins on the 2-DE gels. iTRAQ is a gel-free high throughput method, and it can usually generate several thousands of unique proteins in plants. Based on the new data produced by the two complementary proteomics methods, we have uncovered some novel mechanisms on the ethylene-stimulation of rubber latex production.

Comment 3: Please compare/discuss your work on the phosphorylation of rubber proteins with a recently published related work: Habib, M. A. H., Gan, C. Y., Abdul Latiff, A., & Ismail, M. N. (2018). Unrestrictive identification of post-translational modifications in Hevea brasiliensis latex. Biochemistry and Cell Biology, 96(6), 818-824.

Response: Thank you very much for this suggestion. We have downloaded this newly published paper and cite it in the revised manuscript. In this newly published paper, Habib et al. (2018) collected proteins from the C-serum, lutoids and rubber particle layers of the rubber latex and then fragmented using collision-induced dissociation (CID), higherenergy collisional dissociation (HCD) and electron-transfer dissociation (ETD) activation methods. PEAKS 7 were used to search for unspecified PTMs, followed by analysis through PTMs prediction tools to crosscheck both results. Finally, 73 peptides in 47 proteins from H. brasiliensis protein sequences derived from UniProtKB were identified and predicted to be posttranslationally modified. The peptides with PTMs identified include phosphorylation, lysine acetylation, N-terminal acetylation, hydroxylation, and ubiquitination. This study is set to analyze the peptides in the proteins from different fractions of rubber latex. Our purpose is to determine the detail phosphorylated peptides and phosphorylated amino acids in the REF/SRPP family. Both studies proved that protein phosphorylation is one of important PTMs in the rubber latex, and phosphorylation of some enzymes might be crucial for natural rubber biosynthesis.

Round 2

Reviewer 1 Report

Thank you for your efforts to answer my questions/comments. I think manuscript quality has improved after revision. I still could not agree on the number of phosphopeptides you have identified. If the same peptide is mapped multiple times, that does not increase the number of identified phosphppeptes. So, authors should only report unique phopshoppeptides. For example, if a peptide ABCDEF is detected 5 times, that does not mean 5 peptides were identified. Only 1 peptide is identified with five spectral counts. Please correct this accordingly. You can refer to many published papers how others have reported identified phosphopetides.

I still see many places where English can be improved/need correction.Please read the manuscript carefully for any improvement in grammar and readability. I think this manuscript can be accepted after addressing these issues.

Author Response

Thank you for your helpful suggestion. We have re-counted the number of the phosphopeptides and the phosphorylation amino acids, and only reported the unique phopshoppeptides in the newly revised manuscript. Accordingly, a new Table S2 has been provided in the new supplementary data.

We have further carefully polished the manuscript. The English has been improved and some typographical, grammatical, spelling, vocabulary, and confusing sentences have been corrected in the newly revised manuscript. Therefore, this newly revised version might be more suitable for publication now. We have also submitted a highlighted version showing where revisions have been made in the original manuscript.

Reviewer 2 Report

The authors put sufficient responses to the reviewers' comments.

Author Response

Thank you. We have further carefully polished the manuscript. The English has been improved and some typographical, grammatical, spelling, vocabulary, and confusing sentences have been corrected in the newly revised manuscript. Therefore, this newly revised version might be more suitable for publication now. We have also submitted a highlighted version showing where revisions have been made in the original manuscript.